# Influence of Body Condition Score on Reproductive Recovery in Spring and on Some Hematochemical Parameters in Sarda Breed Sheep

**DOI:** 10.3390/ani15030372

**Published:** 2025-01-28

**Authors:** Sebastiano Luridiana, Mortadha Ouadday, Giovanni Cosso, Vincenzo Carcangiu, Maria Consuelo Mura

**Affiliations:** 1Department of Veterinary Medicine, University of Sassari, Via Vienna 2, 07100 Sassari, Italy; sluridiana@uniss.it (S.L.); vcarcangiu@uniss.it (V.C.); 2National School of Veterinary Medicine, Sidi Thabet, University of Manouba, La Manouba 2010, Tunisia; mortadha.ouadday@gmail.com; 3Agris Sardegna, Bonassai, SS 291, Km 18.600, 07100 Sassari, Italy; gcosso@agrisricerca.it

**Keywords:** body condition score (BCS), male effect, reproductive recovery, Sarda breed, out-of-season ovulation, reproductive performance, blood chemistry parameters, animal welfare

## Abstract

This study explored how the body condition of Sarda dairy sheep affects their reproductive performance in spring, focusing on the benefits of a natural technique called the male effect, which induces ovulation without exogeneous hormones. The research involved 200 healthy sheep (aged 3–5 years) grouped by their body condition score. Sheep with a higher body condition score (3.0–3.5) showed higher fertility, lambed earlier, and performed better overall than those with lower body condition scores (2.0–2.5). Blood tests revealed that sheep in good condition had healthier levels of glucose, cholesterol, and other indicators. These findings highlight the importance of maintaining an optimal body condition through proper nutrition to improve sheep welfare and reproductive success without relying on hormonal treatments. This approach can support more sustainable and ethical farming practices, while helping farmers optimize productivity.

## 1. Introduction

Small ruminants reared in the Mediterranean area exhibit a reproductive seasonality, characterized by recurring physiological alternations between periods of reproductive activity and sexual rest, which are linked to the photoperiod trend [1]. The breeding season for these species typically begins in late summer or early fall, in response to the shortening of the day length, and ends during the late winter or early spring [2].

For dairy sheep raised in the Mediterranean area, lambing at the beginning of autumn is a crucial requirement to ensure a lactation period of six or seven months [3]. The Mediterranean climate is characterized by a mild winter, with rainfall occurring in autumn and spring, and a very dry summer. This climate affects grass growth, which is usually available in the fall and, especially, in the spring. Therefore, out-of-season lambing in autumn is necessary to optimally exploit the herbage growth cycle. To enable autumn lambing, sheep should mate in spring, when they are in anestrus. During the springtime, however, despite the sheep being adequately sensitized to long days, reproductive activity is less stimulable [4].

The reproductive activity of sheep can be stimulated with several methods, as developed in recent decades [5]. Some of these methods involve the administration of exogenous hormones, such as progesterone, prostaglandins, melatonin, etc., which stimulate reproductive activity in various ways [6]. These methods are efficient in controlling the ovarian cycle, allowing for the synchronization of estrus and enabling the execution of artificial insemination, which is crucial for genetic improvement plans [7].

Furthermore, on organic farms, where the administration of certain drugs is not permitted, these treatments cannot be used, and other methods that do not involve the use of hormones, such as flushing (a process that involves increasing sheep nutrition and energy intake before the breeding season starts) and exposure to the male effect, are essential to stimulate reproductive activity [8]. The male effect works by stimulating the hypothalamic–pituitary–gonadal axis, triggering hormonal changes that promote the resumption of ovarian activity and estrus in ewes [9]. These methods are also called “green methods” as they leave no residues in the animals or in their grazing land, ensuring the eco-sustainability of the breeding process [10]. While the use of these methods does not guarantee precise synchronization, they can shorten the lambing period [8]. However, the response to these methods may be influenced by several factors, including inadequate nutrition, a poor BCS, and suboptimal management of the animals before their inclusion in reproductive plans. Additionally, the number of ewes responding to the male effect during the anestrus season is highly variable [11]. One of the factors influencing suboptimal responses to the male effect is the body condition score (BCS), as research has shown that animals with a low BCS can have low reproductive efficiency [12].

BCS has now become recognized as an indicator of the nutritional status of sheep [13]. Several metabolic parameters may also be indicators of an animals’ nutritional status and may influence their reproductive response [14]. In fact, several studies have highlighted the relationship between ovarian performance and the concentrations of glucose and some hormones in sheep with different nutritional levels [15,16,17,18]. Glucose is now known to be the primary energy source and metabolic substrate essential for ovarian and follicle development [19,20]. In addition to glucose, other blood parameters, such as total cholesterol, triglycerides, total protein, and albumin, provide valuable insights into the animal’s nutritional and metabolic status. Cholesterol and triglycerides are crucial as precursors for steroid hormone synthesis, which is directly linked to reproductive activity [14]. Total protein and albumin reflect the protein metabolism and overall nutritional balance of the animals, which can indirectly affect ovarian function and reproductive outcomes [14]. These parameters collectively contribute to ovarian performance and reproductive efficiency in sheep with varying nutritional levels.

However, despite the importance of these parameters in the management of reproduction and nutrition in sheep farming, there are few studies relating BCS, reproductive activity, and blood parameters, particularly in dairy sheep. Considering the importance of having a reliable exogenous hormone-free method to stimulate reproductive activity and to avoid residues in animal products, it is crucial to determine the optimal BCS value or range to achieve efficient reproductive performance. This knowledge would enable a better-informed nutritional status, maintaining organic homeostasis and animal welfare.

Thus, the aims of this research are (1) to analyze the relationship between different BCS levels and reproductive recovery in spring, stimulated by the male effect, in order to identify the optimal BCS value or range for achieving good reproductive activity in Sarda sheep and (2) to investigate the relationship between BCS and some hematochemical parameters in order to maintain organic homeostasis and ensure animal welfare.

## 2. Materials and Methods

### 2.1. Animals and Experimental Design

The animals used in the research were managed and cared for by the veterinary farm, which followed the guidelines of the Animal Welfare Organization and was under the control of the National Health Veterinary Service. The treatments performed on animals were techniques commonly used in sheep farming to improve reproductive performance. The study was conducted in a Sarda sheep farm, located around the 40th parallel (40°46 N 8°30 E), which housed approximately 1000 Sarda sheep, maintained under natural photoperiodic condition since birth. Sardinia features a typical Mediterranean climate characterized by hot, dry summers and mild, wet winters. The region follows a Mediterranean rainfall pattern, with nearly dry summers and rainy autumns, winters, and springs, with annual precipitation generally ranging between 500 and 700 mm. In terms of photoperiod, Sardinia experiences considerable seasonal variation in daylight hours, typical of Mediterranean climates. In mid-May, when the males were introduced (on 15 May), the ewes experienced approximately 14–15 h of daylight. Over the 50-day study period, as the region transitioned from late spring to early summer, daylight hours gradually decreased. From June to September, temperatures in Sardinia typically range between 23 and 28 °C, though days exceeding 30 °C are not uncommon. Sardinia is also a windy island, particularly from October to April, due to the Mistral, a northwesterly wind that brings cool air and sunny days, causing a daily temperature drops of up to 5 °C. Conversely, hot and humid winds like the Sirocco can elevate temperatures to as high as 40 °C.

The Sarda is the principal Italian dairy sheep, known for its high milk production compared to other Italian breeds [21]. In Sardinia, these sheep are primarily reared in semi-extensive conditions, with a feeding strategy that relies mainly on natural pastures. The average milk yield per lactation in autumn-lambing adult ewes of the unselected common population, based on our experience, ranges from 200 to 300 liters but can even exceed 450 liters in subjects selected through genetic improvement plans and properly managed.

During the day, the ewes involved in the study grazed on leguminous grasses such as *Trifolium subterraneum* and *Medicago sativa* and gramineous species such as *Lolium perenne* and *Avena sativa, Hordeum vulgare, Dactylis glomerata*, and others. Before the introduction of the males, the ewes underwent a nutritional management strategy aimed at improving their reproductive performance, commonly known as ‘flushing’. This practice involves increasing the energy intake of the ewes for approximately 3 weeks prior to the introduction of the males to stimulate estrus and improve ovulation rates. The ewes received 400 g of concentrate commercial feed (crude protein 20.4% and 12.5 MJ ME/kg DM) per head daily during milking. At night, the ewes were maintained in the sheepfold, where they received hay (crude protein 11.1% and 7.2 MJ ME/kg DM) and water ad libitum. This regimen was maintained for the duration of the study, with no further supplementation or change in the forage base at grazing.

On 12 March 2022, a total of 200 healthy lactating ewes, aged 3–5 years (average age 3.8 years), and lambed during the previous season between 1 November and 1 December 2021, were selected from the flock. More precisely, 50 ewes with BCS 2.0, 50 with BCS 2.5, 50 with BCS 3.0 and 50 with BCS 3.5 were chosen to build 4 groups. The body weight of the ewes was also recorded at the time of BCS evaluation. Ewes assigned to BCSs had average body weights between 38 and 46 kg, which were considered typical for the Sarda breed at this stage of lactation and body condition. The selection of 50 ewes per group was aimed at ensuring statistical representativeness and balance between the groups, as well as adequate power to detect potential differences in the variables studied. Ewes were selected from a single flock of Sarda sheep with similar genetic backgrounds to ensure that any differences observed in reproductive response were not confounded by genetic factors. The selection process aimed to balance genetic diversity across the BCS groups. No significant differences emerged in the production levels of the ewes within each group or between groups (average daily milk production 1800 ± 180 g/die). The number of the electronic ruminal bolus was recorded for each animal through a specific reader (Allflex RS420) (Allfelx Livestock Intelligence, portable stick reader, Palmerston North, New Zealand) and their respective BCSs were also registered according to the methods of Russel et al. [22]. Briefly, the scores ranged from 1 (emaciated) to 5 (obese) in increments of half-units. Scoring was based on the palpation (by the veterinary team) of the amounts of muscle and fat deposition over and around the vertebrae in the loin region. The engaged ewes were at least in their third lambing, in good health and nutritional condition. The exclusion of ewes at first and second lambing agreed with Mura et al. [8], as they do not exhibit their full reproductive and productive potential. Briefly, in the Sarda breed, the first lambing generally occurs between January and April, but this breed exhibits a high milk yield, leading to poor reproductive activity for approximately 2 months after lambing. As a result, there is a delay in the second lambing compared to adult ewes, so only multiparous ewes were selected for the present study.

Generally, lambs remain with their mothers and suckle for approximately 30 days, during which the ewes are not milked, and their entire milk production is dedicated to the lambs. After weaning, ewes are typically milked twice daily, with lactation lasting approximately 180 days. By late spring or early summer, they are simultaneously dried off due to the onset of the summer drought.

The grouped ewes were separated from the remaining flock and kept isolated for the entire experimental period (about 53 days). More precisely, the 200 ewes were divided into four experimental groups of 50 ewes each, but they were all kept together as a single flock. On May 15, 10 fertile rams (male/female ratio 1/20) were introduced into this combined flock of 200 ewes and remained there for 50 days before being removed. The rams were clinically healthy, aged 4–6 years, with an average BCS ≥ 3.5. Their fertility was proven through previous successful progeny production, and they underwent routine reproductive examinations by the farm veterinarians. The ewes had previously been isolated from the rams for a period of about 6 months. The rams were kept 5 km away from the ewes, ensuring that the females could not be exposed to the rams through sight, sound, or smell during the isolation period. Moreover, each ram was fitted with a marking harness for estrous detection. The number of ewes with ram keel marks were recorded daily (during milking), with the keel color changed every 7 days. Furthermore, the sexual behavior of the rams was observed (by the farm staff who were responsible for the daily management of the animals) during the first few days after their introduction into the groups to confirm readiness for sexual activity. The observed behaviors included anogenital sniffing, nudging, and mounting attempts directed at the ewes. At the time of the males’ introduction, the Sarda ewes were typically in a transitional state, following a period of deep anestrus that generally occurs between February and April.

Several factors, such as climatic conditions, individual animal variability, and the management conditions on the farm, including the level of human intervention, might have contributed to variations in the reproductive response. While the study was designed to minimize these influences, it is important to acknowledge that the results may not be fully generalizable to other production systems or breeds.

BCS was assessed by manually evaluating the degree of muscling and fat deposits in the loin region on 12 May (which was the time of the groups’ formation, three days before the rams’ introduction), as well as at 15, 30, and 50 days after the males’ introduction. The same technicians consistently performed the BCS evaluations. For each assessment, BCS 2.0 was assigned to ewes with a slight fat coverage and detectable but not sharp vertebrae and pelvic bones; BCS 2.5 represented a moderate fat coverage where the lumbar vertebrae were rounded and the pelvic bones less prominent; BCS 3.0 corresponded to an optimal condition with good fat coverage over the vertebrae, which could be felt with pressure but not sharply defined; and BCS 3.5 was associated with a higher fat deposition that made vertebrae and pelvic bones difficult to distinguish even with firm palpation. On those same dates, blood samples were collected from the jugular vein at 07.00 a.m., using vacuum tubes containing lithium heparin. The blood was transported in refrigerated containers at 5 °C to the laboratories of the Department of Veterinary Medicine of Sassari within one hour and centrifuged. The plasma was aliquoted and frozen at −20 °C until analysis.

### 2.2. Blood Chemistry Analysis and Reproductive Data Collection

Glucose, total cholesterol, triglycerides, total protein, and albumin blood concentrations were determined using colorimetric methods with commercial kits (Sentinel Diagnostic, Milan, Italy) and by an automated spectrophotometer (Vitalab 200R, Vital Scientific, Dieren, The Netherlands).

Gestation was diagnosed between 45 and 90 days after mating by performing a transabdominal ultrasonography examination using Esaote Piemedical Tringa linear equipment (Esaote Europe B.V., Maastricht, The Netherlands) provided with a 5.0–7.5 MHz multiple-frequency linear probe. The lambing dates and number of lambs born were recorded from 150 to 200 days after the males’ introduction. From the recorded reproductive data, the fertility rate (number of ewes that lambed per ewes exposed to the male), litter size (number of lambs born per ewe that lambed), and the distance in days from male introduction to lambing (DRIL) were calculated. All reproductive parameters were calculated at the end of the lambing period, which occurred between 12 October and 1 December (150–200 days after the introduction of the rams on 15 May), based on the number of successfully pregnant animals, as confirmed by ultrasonography and subsequent lambing, relative to the total number of animals bred.

### 2.3. Statistical Analysis

Statistical analysis was performed using R statistical software (version 4.4.0, R Core Team (2024) [23].

All the results were considered statistically significant when the *p*-value was below 0.05. The differences in fertility between the four groups were tested using the Chi-square test, while the differences in litter size and DRIL were analyzed using the following model:Y_ijkm_ = µ + B_i_ + P_j_ + A_k_ + r_m_ + e_ijkm_
(1)
where Y_ijkm_ is the trait measured for each animal studied (DRIL or litter size), B_i_ is the fixed effect of the BCS group (4 levels: 2.0, 2.5, 3.0, 3.5), P_j_ is the covariate for productive level (continuous variable, e.g., liters of milk per lactation), A_k_ is the fixed effect of the age (2 levels: 3 or 4 years old), r_m_ is the random effect of the rams (10 rams), and _ejkm_ is the random residual effect for each observation. The least square means ± SEM were used to express DRIL and litter size. Tukey’s method was used to perform multiple comparisons of the least square means.

The differences in blood glucose, total cholesterol, and total proteins levels between the four groups were statistically analyzed using linear mixed-model regression (REML) implemented in the lme4_4.0.2 R Package [24], where the BCS (4 levels), period of sampling (4 levels), productive level (continuous covariate), and age (2 levels) were set as fixed effects, and *e* was the random residual effect of each observation.

## 3. Results

The BCS remained largely constant across the four assessments performed (T0, T15, T30, and T50) (Table 1), showing only a slight growth, in all groups, without statistically significant differences among them.

The number of ewes diagnosed as pregnant and the number of ewes that lambed showed a difference of approximately 2%. The fertility rate was higher (*p* < 0.01) in sheep with a BCS of 3.0 and 3.5 than in the other two groups (Table 2). Furthermore, the group with a BCS of 2.5 had a higher rate of fertility than the group with a BCS of 2.0. Conversely, the incidence of non-pregnant ewes was much higher in the groups with a BCS of 2.0 and 2.5 than in the other two groups (*p* < 0.01) (Table 2).

Ewes with a BCS of 3 and 3.5 lambed approximately 15 and 25 days earlier on average, respectively, compared to the other two groups (*p* < 0.01) (Table 2). Although the litter size was higher in ewes with the highest BCS, the differences were not statistically significant (Table 2).

The lambing distribution indicates that ewes with the highest BCS lambed considerably earlier than those with the lowest BCS. Specifically, the peak of lambing occurred at 170 days from the males’ introduction for ewes with a BCS of 3.5, at 180 days for those with a BCS of 3.0, at 190 days for those with a BCS of 2.5, and at 200 days for those with a BCS of 2.0 (Figure 1).

Moreover, the cumulative lambing trend showed that the group with a BCS of 3.5 had the best reproductive performances at 160, 170, and 180 days from ram introduction compared to the other three groups (*p* < 0.01) (Figure 2). Additionally, the two groups with the highest BCS had a greater number of lambed ewes at 160, 170, 180, and 190 days after the males’ introduction compared to the groups with the lowest BCS (*p* < 0.01) (Figure 2).

In the evaluation of reproductive performance, returns to estrus were observed in certain ewes: six ewes in the BCS 2.0 group, four in the BCS 2.5 group, one in the BCS 3.0 group, and none in the BCS 3.5 group. These differences in returns to estrus contributed to the variations in DRIL, as shown in Table 2.

Blood glucose concentrations showed a consistent trend across all groups throughout the observation period. In all four samples collected, the highest levels of this parameter were recorded in animals with a higher BCS (3.5 and 3.0) compared to the groups with the lowest BCS (2.5 and 2.0) (*p* < 0.05) (Table 3).

Total cholesterol levels showed the highest concentrations in the two groups with the highest BCS compared to the other two groups across all the samples taken (*p* < 0.05) (Table 4).

No differences were found within the groups for this parameter during the samplings. Total cholesterol and triglycerides showed the same increasing trend during the observations in all groups, but without any statistically significant differences. The two groups with the highest BCS exhibited higher blood triglyceride concentrations than the other two groups (*p* < 0.05) (Table 5).

The concentrations of total proteins and albumins remained constant, without showing differences between samples and between groups (Table 6 and Table 7).

## 4. Discussion

The main objective of this study was to investigate the impact of BCS on reproductive performance and metabolic parameters in ewes. The obtained results clearly demonstrated that ewes with a higher BCS (3.0 and 3.5) exhibited significantly improved fertility rates, earlier lambing, and a higher number of ewes lambing earlier compared to those with a lower BCS (2.0 and 2.5). Ewes with a BCS of 3 and 3.5 lambed approximately 20 days earlier on average, aligning with previous studies, such as Forcada et al. [25], who observed a reduction in the duration of seasonal anestrus in ewes with a high BCS (2.9) compared to those with a lower BCS (2.3). These findings suggest that nutritional status, reflected by BCS, can be considered a significant factor in regulating reproductive activity in sheep, with higher BCS levels indicating a better nutritional status and improved reproductive outcomes [26]. This study also highlighted a consistent trend of higher blood glucose, cholesterol, and triglyceride levels in ewes with a higher BCS, suggesting a potential link between metabolic status and reproductive success. Given its association with both reproductive performance and overall health, BCS can serve as a valuable health management tool, providing insights into the animal’s energy reserves and welfare status [14,27,28].

The returns to estrus observed in the groups suggest that BCS may influence the onset of estrus after ram introduction. Ewes with a lower BCS (2.0 and 2.5) exhibited a higher number of returns to estrus, indicating a delayed or suboptimal reproductive recovery compared to ewes with a higher BCS (3.0 and 3.5). These findings further support the importance of maintaining an optimal BCS to enhance reproductive performance and minimize estrus irregularities before pregnancy, aligning with the findings of Luridiana et al. [12], who emphasized the critical role of nutritional status and BCS in efficient reproductive recovery.

Various studies had shown that short-term supplementation before the mating period may boost reproductive efficiency in Mediterranean sheep-breeding systems [29]. Undoubtedly, the quality of the pasture influences not only the individual BCS but also the reproductive performance in dairy sheep [30,31]. However, on pasture, the feed intake of sheep is not evenly distributed, as each sheep consumes a different amount of forage based on various factors, such as individual preference for certain plants. This study was conducted in a private flock, following traditional farm management in a semi-extensive breeding system, which made it impossible to monitor precisely how much grass or forage each animal consumed. Consequently, the individual feed intake on pasture remained variable and uncontrolled, although the entire flock grazed in the same pasture under the same conditions. The only consistent feeding factor for each sheep was the supplementation in the milking parlor, where each ewe received 400 g of concentrate feed, which constituted a standardized and controlled daily feeding intake.

In addition to genetics and nutrition, factors such as parity, the season of lambing, milking frequency, and the stage of lactation also play a role in influencing milk composition and yield in ewes, leading to the highest daily milk yield typically achieved during the second and third lactation until the sixth lactation (around six years of age) [32].

Although the relationship between BCS and the milk production of dairy animals has been mostly determined for cows, in sheep this aspect shows mixed results [33]. While it is generally expected that ewes with a lower BCS produce less milk, studies have been inconsistent. Indeed, some studies found no impact of BCS on milk production during breeding, mid-pregnancy, or late pregnancy, while others reported a positive relationship between BCS in late pregnancy and milk production in different sheep breeds [33]. During early lactation, ewes can mobilize their body fat and protein reserves to produce up to one-third of their milk [34]. Ewes with a higher BCS tend to produce more milk if they lose body weight during lactation.

However, when ewes had access to adequate feed, BCS does not seem to affect milk yield. At lower feeding levels, ewes with less body fat were more affected in terms of milk production compared to those with greater fat reserves [34]. Interestingly, in Sarda ewes, a higher BCS during mid-lactation was associated with lower milk production, potentially due to excessive fat compressing the rumen, limiting feed intake [35]. The productive attitude of the Sarda breed and its specific traits appear to determine important differences in the response to BCS variations. It is worth noting that the most appreciated characteristics of the Sarda sheep are its small size, frugality, and rusticity which make it generally well-suited to extensive and sometimes quite challenging breeding conditions, while still maintaining good production levels.

Some research has shown that increased feed intake can stimulate multiple ovulations in females and increased sperm production in males [36,37]. For example, dietary modifications to increase the availability of amino acids can positively influence the quality of oocytes during antral follicle development and have a favorable influence on fetal and placental metabolism [38]. However, increases in nutritional intake in sheep with a BCS greater than 3.0 led to only moderate improvements in reproductive performance [39]. Previous studies have highlighted that metabolic balance signals influence the activity of GnRH neurons in the hypothalamus and therefore the functionality of the gonads [40]. The influence of metabolic signals, particularly insulin, glucose, and leptin, on GnRH neurons is now recognized as playing an essential role in regulating hypothalamic activity in sheep [40,41]. The influence of metabolic parameters was further elucidated through the longitudinal trends observed in this study. Specifically, the gradual increase in glucose and cholesterol levels in higher BCS groups mirrored their enhanced reproductive performance over time. This finding aligns with the hypothesis that metabolic energy reserves play a pivotal role in follicular development and ovulation. Furthermore, while triglyceride levels showed a consistent association with BCS, their specific longitudinal variations suggest a role in steroidogenesis, potentially mediated by unsaturated fatty acids. The role of leptin, which is modulated by BCS and interacts with hypothalamic–pituitary–ovarian pathways, further underscores the complex metabolic regulation of reproduction. These dynamic metabolic profiles emphasize the necessity of considering both static and temporal factors when evaluating reproductive outcomes. The quantity and quality of feed consumed, and the level of the animal’s body reserves, influence the circulating concentrations of insulin and leptin, which in turn determine control over the pulsatile secretion of GnRH and LH [42].

Furthermore, receptors for insulin and glucose have been identified in the ovarian follicle, both in the granulosa and theca cells. These receptors are responsible for the activation of various intracellular kinases [19,43,44,45], which is essential for the life of the follicle and oocyte. The inhibition of these kinases can lead to the non-proliferation of granulosa cells or premature death of the follicle [46,47]. Therefore, the insulin–glucose system could have crucial functions in maintaining the health and cellular integrity of follicular cells, as in other cells of the body. Furthermore, in vitro studies have suggested that the insulin–glucose system may have specific actions on granulosa and theca cells, potentially even exhibiting direct gonadotropic effects [48]. In support of this, granulosa and theca cells have been observed to proliferate in an insulin dose-dependent manner [49,50].

Leptin and its mRNA have been identified in the theca and oocyte, while in granulosa cells, leptin is only weakly expressed [51]. Furthermore, leptin receptors have been characterized in granulosa cells, theca cells, and the oocyte [52,53]. These findings suggest that leptin, in addition to its endocrine functions, may also have paracrine and possibly autocrine functions in the follicle. Furthermore, some studies suggest that leptin administration stimulates folliculogenesis in sheep [52,54]. The administration of leptin into the ovarian artery caused a decrease in the production of estradiol from the ovary, a finding that was confirmed in vitro using granulosa cells [54,55]. The inhibition of leptin on the secretion of estradiol appears to involve the insulin-like growth factor (IGF) system, as leptin can inhibit the effect played by IGF on the production of estradiol stimulated by FSH in granulosa cells [55,56]. Consequently, leptin plays an important role in steroidogenesis at both the granulosa and theca cell levels [57]. This suggests that a higher BCS, through metabolic signals, could have influenced the secretion of GnRH and LH at the hypothalamic–pituitary–ovarian level, ultimately leading to the better reproductive activity, stimulated by the male effect, observed in the groups with a higher BCS (3.0 and 3.5) compared to the groups with a lower BCS (2.0 and 2.5).

BCS levels also influenced the concentrations of different metabolic parameters, even if they always remained within the physiological ranges of the species and breed [58]. Animals with a higher BCS showed higher levels of blood glucose, total cholesterol, and triglycerides. The findings regarding blood glucose and triglyceride levels aligned with what was found by Caldeira et al. [14], who reported a direct relationship between these parameters and BCS. However, the present study observed an increasing trend in total cholesterol with a higher BCS, whereas Caldeira et al. [14] did not explicitly report trends for total cholesterol. Instead, their study focused on total lipids, which showed no significant differences between groups. This discrepancy could be attributed to differences in physiological states, diets, and experimental conditions.

The higher blood glucose levels observed in this study are related to the BCS, likely reflecting the improved nutritional and metabolic state of the animals with greater BCS. This elevated glucose may have acted as a positive signal, contributing to the better reproductive activity recorded in animals with a higher BCS. Glucose is an essential substrate for metabolism, and its actions at the cellular level activate several physiological mechanisms. Interestingly, ovarian venous blood has lower glucose concentrations than carotid arterial blood, indicating that the ovary actively captures glucose from the blood circulation [19]. The presence of glucose transport channels, GLUT1 and GLUT4, in the granulosa and theca cells of sheep and cattle [59,60] further highlights the important role of insulin in regulating glucose uptake at the ovarian level. Therefore, the higher glucose levels in animals with BCS 3.0 and 3.5 may have facilitated improved follicular development and, consequently their enhanced reproductive performances.

The higher levels of blood lipids found in ewes with the highest BCS may also have influenced their reproductive activity. In fact, unsaturated fatty acids have been found to play a role in regulating female reproduction. In cattle, the administration of long-chain polyunsaturated fatty acids (PUFAs) influenced oocyte quality and improved folliculogenesis [61]. Therefore, it could be hypothesized that triglycerides or total cholesterol levels may have also improved the entire process of folliculogenesis, potentially influencing steroidogenesis.

Blood protein and albumin concentrations remained unaffected by different BCS levels. These parameters are physiologically quite stable, and only significant changes in nutritional state or pathological conditions can lead to alterations in their homeostasis. Although it is possible that subtle variations in these parameters might have been observed with a larger sample size, an extended observation period, or the analysis of additional hormones, this did not affect the overall objective of the research, which was to explore the relationship between BCS and reproductive performance.

## 5. Conclusions

In conclusion, the results of this study effectively addressed the objectives, highlighting the importance of BCS as a key management tool for dairy sheep producers. This study underscores the significance of BCS as an indicator of the energetic nutritional status of Sarda breed sheep, demonstrating its impact on reproductive activity and certain hematochemical parameters, all while maintaining milk yield. Notably, BCS levels of 3.0 and 3.5 were associated with enhanced fertility and quicker reproductive recovery in the spring, highlighting the potential for improved reproductive performance through effective nutritional strategies. Moreover, sheep within these optimal BCS ranges exhibited a more balanced metabolic state, reflected in favorable blood chemistry parameters. This reinforces the critical role of understanding BCS in relation to animal performance traits. By adopting targeted nutritional management strategies adapted to the different stages of production, sheep producers can enhance productivity and ensure the welfare of their animals. This study emphasizes the importance of BCS monitoring as a tool for precise nutritional management, which is essential for optimizing reproductive efficiency. It supports the implementation of hormone-free breeding programs for small ruminants, promoting a more sustainable and welfare-oriented approach to sheep production.

Further studies should explore the long-term effects of BCS on overall animal health and productivity. Investigating the potential for improving reproductive performance through more precise nutritional interventions, as well as examining the influence of different BCS ranges on the sustainability and welfare of sheep, could provide valuable insights for enhancing farm management practices. Additionally, future studies on the relationship between BCS and hormones regulating reproductive and metabolic activity could certainly improve the understanding of the mechanisms underlying the regulation of reproductive resumption and an optimal metabolic condition.

## Figures and Tables

**Figure 1 animals-15-00372-f001:**
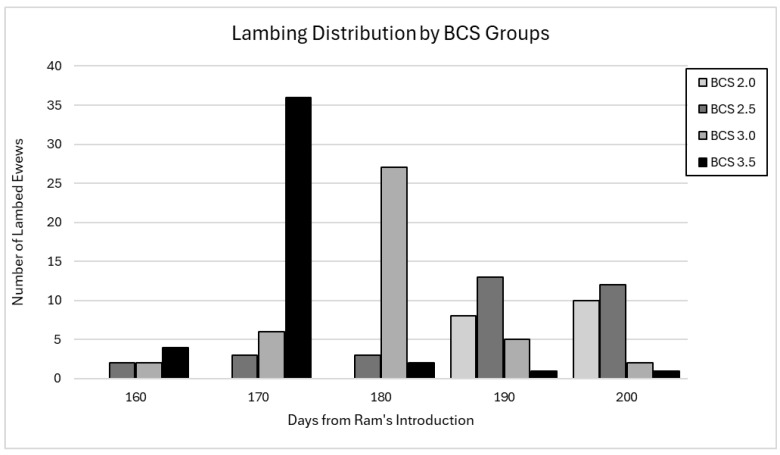
Distribution of lambing every 10 days for the four BCS groups, calculated starting from the 150th day after ram introduction. The bars represent the number of ewes lambing for each BCS group (2.0, 2.5, 3.0, and 3.5).

**Figure 2 animals-15-00372-f002:**
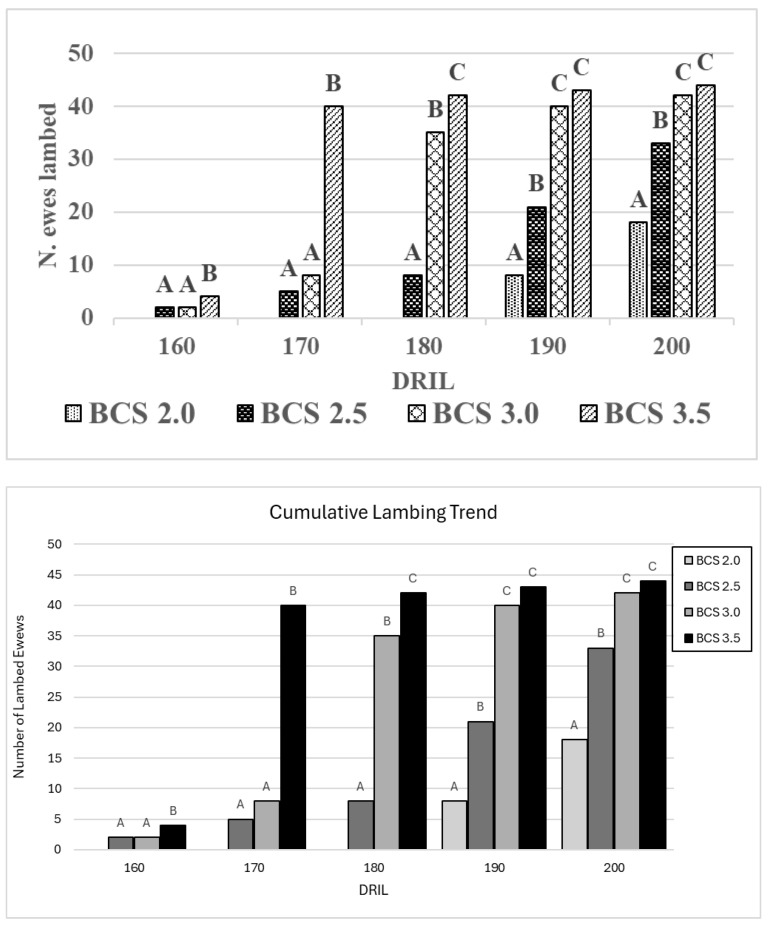
Cumulative lambing trend, every 10 days, starting from day 150 after ram introduction in the four studied groups (A, B, C = *p* < 0.01).

**Table 1 animals-15-00372-t001:** Average body condition score (BCS) ± standard deviation (SD) recorded at 0, 15, 30, 50 days after ram introduction in the four studied groups.

Groups	BCS T0	BCS T15	BCS T30	BCS T50
BCS 2.0	2.0 ± 0.00	2.07 ± 0.18	2.18 ± 0.14	2.20 ± 0.21
BCS 2.5	2.5 ± 0.00	2.52 ± 0.17	2.54 ± 0.16	2.66 ± 0.20
BCS 3.0	3.0 ± 0.00	3.08 ± 0.15	3.17 ± 0.19	3.19 ± 0.18
BCS 3.5	3.5 ± 0.00	3.52 ± 0.19	3.52 ± 0.21	3.45 ± 0.15

**Table 2 animals-15-00372-t002:** Fertility rate, number of lambed and empty ewes, litter size, and DRIL (distance in days from ram introduction to lambing) in four studied groups with different body condition score (BCS) in Sarda sheep.

Groups	Fertility Rate	Lambed Ewes	Empty Ewes	Litter Size	DRIL
BCS 2.0	36%	18 ^A^	32 ^C^	1.22 ± 0.04	192.20 ± 4.60 ^C^
BCS 2.5	66%	33 ^B^	17 ^B^	1.18 ± 0.03	190.23 ± 4.38 ^C^
BCS 3.0	84%	42 ^C^	8 ^A^	1.31 ± 0.05	175.05 ± 6.16 ^B^
BCS 3.5	88%	44 ^C^	6 ^A^	1.33 ± 0.04	166.76 ± 3.63 ^A^

Significant statistical differences are highlighted by the uppercase letters within the column (A, B, C = *p* < 0.01).

**Table 3 animals-15-00372-t003:** Mean ± s.d. of blood glucose values (mg/dL) at 0, 15, 30, 50 days after ram introduction in the four groups.

	Time of Sampling for Blood Glucose Levels (mg/dL)			
Groups	T0	T25	T30	T50
BCS 2.0	47.05 ± 4.66 ^a^	46.08 ± 3.59 ^a^	46.79 ± 4.37 ^a^	48.45 ± 4.85 ^a^
BCS 2.5	54.88 ± 8.32 ^a^	53.19 ± 6.12 ^a^	53.48 ± 6.17 ^a^	54.37 ± 6.52 ^a^
BCS 3.0	59.33 ± 5.28 ^b^	58.50 ± 4.72 ^b^	58.98 ± 5.14 ^b^	60.12 ± 5.38 ^b^
BCS 3.5	66.03 ± 8.58 ^b^	64.95 ± 5.80 ^b^	65.04 ± 5.32 ^b^	65.49 ± 5.85 ^b^

Significant statistical differences are highlighted by the lowercase letters within the column (a, b, = *p* < 0.05).

**Table 4 animals-15-00372-t004:** Mean ± s.d. of blood total cholesterol values (mg/dL) at 0, 15, 30, 50 days after ram introduction in the four groups.

	Time of Sampling for Blood Cholesterol Levels (mg/dL)			
Groups	T0	T25	T30	T50
BCS 2.0	43.88 ± 7.21 ^a^	45.08 ± 4.46 ^a^	46.82 ± 4.45 ^a^	47.12 ± 5.38 ^a^
BCS 2.5	51.17 ± 7.53 ^a^	51.95 ± 6.78 ^a^	53.07 ± 6.26 ^a^	53.74 ± 6.12 ^a^
BCS 3.0	61.88 ± 7.83 ^b^	61.90 ± 6.01 ^b^	63.65 ± 5.54 ^b^	64.63 ± 6.71 ^b^
BCS 3.5	66.47 ± 10.62 ^b^	66.16 ± 7.37 ^b^	67.56 ± 7.30 ^b^	68.34 ± 6.94 ^b^

Significant statistical differences are highlighted by the lowercase letters within the column (a, b, = *p* < 0.05).

**Table 5 animals-15-00372-t005:** Mean ± s.d. of blood triglyceride values (mg/dL) at 0, 15, 30, 50 days after ram introduction in the four studied groups.

	Time of Sampling for Blood Triglycerides (mg/dL)			
Groups	T0	T25	T30	T50
BCS 2.0	21.92 ± 5.56 ^a^	21.77 ± 4.89 ^a^	23.87 ± 5.31 ^a^	27.12 ± 5.38 ^a^
BCS 2.5	25.54 ± 6.04 ^a^	26.83 ± 4.89 ^a^	29.05 ± 5.98 ^a^	28.74 ± 6.37 ^a^
BCS 3.0	33.34 ± 8.06 ^b^	33.68 ± 5.99 ^b^	35.13 ± 9.25 ^b^	38.51 ± 7.84 ^b^
BCS 3.5	39.27 ± 8.13 ^b^	36.51 ± 6.56 ^b^	36.27 ± 5.99 ^b^	38.34 ± 6.38 ^b^

Significant statistical differences are highlighted by the lowercase letters within the column (a, b, = *p* < 0.05).

**Table 6 animals-15-00372-t006:** Mean ± s.d. of blood total protein values (g/dL) at 0, 15, 30, 50 days after ram introduction in the four studied groups.

	Time of Sampling for Total Proteins (g/dL)			
Groups	T0	T25	T30	T50
BCS 2.0	6.92 ± 0.10	7.21 ± 0.15	7.14 ± 0.23	7.12 ± 0.18
BCS 2.5	7.02 ± 0.12	7.17 ± 0.19	7.14 ± 0.18	7.23 ± 0.20
BCS 3.0	7.18 ± 0.19	7.11 ± 0.19	7.15 ± 0.15	7.51 ± 0.18
BCS 3.5	7.29 ± 0.19	7.11 ± 0.17	7.13 ± 0.19	7.37 ± 0.21

**Table 7 animals-15-00372-t007:** Mean ± s.d. of blood albumin values (g/dL) at 0, 15, 30, 50 days after ram introduction in the four studied groups.

	Time of Sampling for Total Albumin (g/dL)			
Groups	T0	T25	T30	T50
BCS 2.0	3.23 ± 0.12	3.19 ± 0.13	3.27 ± 0.15	3.27 ± 0.17
BCS 2.5	3.18 ± 0.15	3.25 ± 0.18	3.22 ± 0.16	3.29 ± 0.13
BCS 3.0	3.56 ± 0.17	3.48 ± 0.19	3.52 ± 0.15	3.51 ± 0.19
BCS 3.5	3.61 ± 0.20	3.71 ± 0.14	3.65 ± 0.18	3.78 ± 0.14

Significant statistical differences are highlighted by the lowercase letters within the column (a, b, = *p* < 0.05).

## Data Availability

The original contributions presented in the study are included in the article, further inquiries can be directed to the corresponding author.

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
