# Peer review of "Influence of Body Condition Score on Reproductive Recovery in Spring and on Some Hematochemical Parameters in Sarda Breed Sheep"

_animals, 2025, doi:10.3390/ani15030372_

Round 1
Reviewer 1 Report
Comments and Suggestions for Authors
This is an interesting manuscript aimed to study the relationship between body condition score (BCS), reproductive recovery and hematological parameters in Sarda breed ewes. Introduction section is comprehensible and describes the study in the broad context highlighting its importance, but Materials and Methods section would improve by adding more detailed information about animals, BCS and materials (i.e., kits). I also suggest considering some minor comments in Results, Discussion and Conclusions sections.
- Abstract:
1) Lines 23-24: Please consider the second objective described at the end of the Introduction section.
2) Line 26: It would be very useful to briefly describe how the BCS was measured because it was the central measurement of the study.
- Materials and Methods:
1) Lines 103-132: Very long paragraph, I suggest to separate it in 2 or 3 shorter paragraphs.
2) Line 115: I suggest providing a more detailed description of the ewes (i.e., age, physiological state, number of births, etc.).
3) Line 115: What was the reason for selecting 50 females from each BCS group?
4) Line 123: I suggest providing a more detailed description of the rams (i.e., age, BCS, fertility tests, etc.).
5) Line 133: It is important to describe the technique used to estimate the degree of BCS each time it was assessed (i.e., visual observation, palpation, etc.).
6) Line 133: It is necessary to describe the BCS scale used each time it was assessed. That is, define when a BCS record was assigned equal to 1, equal to 2, and so on.
7) Lines 142-144: Please provide the name of each of the kits used, as well as the manufacturer, for each of the blood parameters evaluated.
8) Line 150: When was the fertility rate calculated?
- Results:
1) Line 184 – Table 1: Are these BCS records average values? If so, please mention it.
2) Line 187: It should be “p” instead of “P”; please check throughout the manuscript.
- Discussion:
1) Line 260: I suggest mentioning briefly in the first paragraph how the main results of the study helped to accomplish the objective.
2) Line 268: Pleas remove the words “(Bronson 1998)”.
3) Lines 292-315: Very long paragraph, I suggest to separate in 2 or 3 shorter paragraphs.
4) Line 383: I suggest mentioning at the end of this section if there were some limitations of the study to meet the objective of the research.
- Conclusions:
1) I suggest describe clearly, at the beginning of this section, whether the results obtained were sufficient to meet the objectives of the study.
2) I suggest including some recommendations for a further studies.
- References:
1) The name of the authors must be separated by a semicolon sign.
Author Response
This is an interesting manuscript aimed to study the relationship between body condition score (BCS), reproductive recovery and hematological parameters in Sarda breed ewes. Introduction section is comprehensible and describes the study in the broad context highlighting its importance, but Materials and Methods section would improve by adding more detailed information about animals, BCS and materials (i.e., kits). I also suggest considering some minor comments in Results, Discussion and Conclusions sections.
Response:
We sincerely thank you for your thoughtful and constructive comments of our manuscript. We have carefully addressed your suggestion to improve the quality of the manuscript as outlined below.
- Abstract:
1) Lines 23-24: Please consider the second objective described at the end of the Introduction section.
Response: the second objective mentioned at the end of the Introduction has now been added to the Abstract, as suggested.
2) Line 26: It would be very useful to briefly describe how the BCS was measured because it was the central measurement of the study.
Response: a brief description of the method used to measure BCS has been added.
- Materials and Methods:
1) Lines 103-132: Very long paragraph, I suggest to separate it in 2 or 3 shorter paragraphs. Response: the lengthy paragraph has been divided into three shorter paragraphs to improve readability.
2) Line 115: I suggest providing a more detailed description of the ewes (i.e., age, physiological state, number of births, etc.).
Response: A more detailed description of the ewes has been added.
3) Line 115: What was the reason for selecting 50 females from each BCS group?
Response: we have explained the rationale for selecting 50 females in each BCS group. The choice was made to ensure statistical representativeness and balance between groups.
4) Line 123: I suggest providing a more detailed description of the rams (i.e., age, BCS, fertility tests, etc.).
Response: additional details about the rams have been provided.
5) Line 133: It is important to describe the technique used to estimate the degree of BCS each time it was assessed (i.e., visual observation, palpation, etc.).
Response: The technique used to estimate BCS has been further specified for each evaluation made.
6) Line 133: It is necessary to describe the BCS scale used each time it was assessed. That is, define when a BCS record was assigned equal to 1, equal to 2, and so on.
Response: The scale and criteria for assessing each score are now explicitly defined.
7) Lines 142-144: Please provide the name of each of the kits used, as well as the manufacturer, for each of the blood parameters evaluated.
Response: the name of each kit used, along with the manufacturer, for the blood parameters evaluated in the revised manuscript have been added.
8) Line 150: When was the fertility rate calculated?
Response: more details about the reproductive parameters calculation have been added.
- Results:
1) Line 184 – Table 1: Are these BCS records average values? If so, please mention it.
Response: We confirm that the BCS records in Table 1 are indeed average values (mean ± standard deviation). To address your comment, we have updated the table title to explicitly mention this information.
2) Line 187: It should be “p” instead of “P”; please check throughout the manuscript.
Response: We have carefully reviewed the manuscript and replaced all occurrences of “P” with the correct lowercase “p” where applicable to ensure consistency with scientific conventions.
- Discussion:
1) Line 260: I suggest mentioning briefly in the first paragraph how the main results of the study helped to accomplish the objective.
Response: Thank you for your suggestion. We have revised the first paragraph of the discussion to briefly mention how the main results of the study contributed to achieving the objective.
2) Line 268: Pleas remove the words “(Bronson 1998)”.
Response: “(Bronson 1998)” removed.
3) Lines 292-315: Very long paragraph, I suggest to separate in 2 or 3 shorter paragraphs.
Response: the paragraph has been divided into three shorter paragraphs to improve readability.
4) Line 383: I suggest mentioning at the end of this section if there were some limitations of the study to meet the objective of the research.
Response: We have added a brief mention in the discussion regarding the potential limitations of the study.
- Conclusions:
1) I suggest describe clearly, at the beginning of this section, whether the results obtained were sufficient to meet the objectives of the study.
Response: a clear statement at the beginning of the conclusion has been added, to indicate that the results obtained in this study effectively addressed the objectives set out.
2) I suggest including some recommendations for a further studies.
Response: a brief paragraph has been added at the end of the conclusion with recommendations for further studies, based on your suggestion.
- References:
1) The name of the authors must be separated by a semicolon sign.
Response: done
Reviewer 2 Report
Comments and Suggestions for Authors
The study is not new, as these results have been reported since the 1990s, as shown by the literature cited, which is mostly very old. It would be expected that solutions would be suggested to situations such as the uneven grazing habits of sheep or local food supplements that, while being inexpensive, would remedy the situation in this region. Indeed, there are countless marginal areas in many parts of the world where it is not possible for females to maintain a body condition score of 3.0-3.5, and therefore it is necessary to find alternatives for feeding management and reproduction control.
Being a topic that has already been widely studied, it would be expected that the current study would be more complete and that hormone levels such as leptin, IGF-1, and insulin would be reported.
The study covers the effect of BCS on females subjected to the male effect but does not mention the physiology of such Effect (its foundations and which hormones are activated).
Materials and methods
• Was the herd in transition, or in deep anestrus?
• Complete coordinates are missing: longitude, altitude
• Describe: photoperiod of the region (and daylight hours during the study period)
• Maximum and minimum temperatures
• Mention the name of the grasses and shrubs that the sheep consumed.
• Were the groups separated? At what distance?
• Management of the males: did they alternate between groups? What body condition did they have?
• Was their diet not increased over time even when the sheep were pregnant?
• In the Materials and Methods they do not clearly describe that the females were milking (how many days does the milking period or the suckling period of the calves last?) and what their production was, although in lines 310-312 they talk about not finding a difference in production between groups but they do not report that result.
• Why is there such a difference in BCS in the same herd? Grazing habits?, hierarchies, less time after weaning their calves?
• A significant number of twins were not recorded, not even in the high BCS groups?
• It is desirable to report the data of the Pregnancy Diagnosis at 45 and 90 days to have a clearer idea of ​​some reproductive parameters.
• Reporting the levels of insulin, leptin IGF-1 would support the results of this study
• Why was prolificacy low in all groups?
It is desirable to include if there are records of:
Return to estrus
Ovulatory rate,
Pregnant females
Females that aborted (in which period; 1st, 2nd or 3rd third of gestation).
Weight of the offspring
Weight of the litter
Types of birth (single, twins, triplets) between groups
DISCUSION
They discuss a very extensive paragraph (lines 292-315) on milk production and do not report this parameter
Table 2. Empty: how many days into the study?
Table 6. It would be appropriate to refer to the result (very outdated) of 68.34 in group BCS 3.5, T50
Miscellaneous
Line
26 ”The male effect is commonly used in sheep farming as an alternative to hormonal treatments, but its success can be influenced by factors such as BCS”. Missing bibliographical reference
15 and 83 Hormone free (are they referring to exogenous hormones?)151 meaning of DRIL
268. Says Bronson (1998) (24)
352 The surname is Viñoles
The identification of the bars in figure 1 is not clearly visible (next to line 219).
Author Response
Rev2
The study is not new, as these results have been reported since the 1990s, as shown by the literature cited, which is mostly very old. It would be expected that solutions would be suggested to situations such as the uneven grazing habits of sheep or local food supplements that, while being inexpensive, would remedy the situation in this region. Indeed, there are countless marginal areas in many parts of the world where it is not possible for females to maintain a body condition score of 3.0-3.5, and therefore it is necessary to find alternatives for feeding management and reproduction control.
Being a topic that has already been widely studied, it would be expected that the current study would be more complete and that hormone levels such as leptin, IGF-1, and insulin would be reported.
The study covers the effect of BCS on females subjected to the male effect but does not mention the physiology of such Effect (its foundations and which hormones are activated).
Response: Thank you for your thorough review of our manuscript. We acknowledge that the topic is not new, and that these aspects have been reported since the 1990s. However, our goal was to offer new insights specific to a Mediterranean region with a rich livestock heritage that has not been widely studied at the international level. This focus on a local context was aimed at providing valuable perspectives that could inform dairy sheep farming in other areas around the world.
Regarding practical solutions such as feeding management and reproduction control in females with a body condition score lower than 3.0-3.5, we agree with your point about the importance of finding cost-effective alternatives. However, our study specifically targeted the relationship between BCS and reproduction, leaving room for future research to address these practical aspects more comprehensively.
Finally, concerning your suggestion to include hormone levels such as leptin, IGF-1, and insulin, we appreciate this valuable input. While we agree that these measurements could enrich the study, our focus was on more accessible and measurable parameters for the contexts we work in. Similarly, regarding the absence of a detailed explanation of the physiological mechanisms underlying the male effect, we appreciate your observation. We have added some indications in the introduction, along with a relevant bibliographic reference in the References section. Consequently, all other references have been adjusted accordingly. We hope that by making these additions, we have further clarified our goal: to explore the relationship between BCS and reproductive performance, with a focus on practical, observable parameters for farm management.
In the future, we aim to expand our investigation to include analyses of additional hormonal and metabolic parameters. This will enable us to provide more precise recommendations to professionals working in the field of reproductive management in sheep farming.
Thank you again for your constructive feedback, and we hope this clarifies our approach.
Materials and methods
• Was the herd in transition, or in deep anestrus?
Response: In the case of the Sarda sheep, the period of deep anestrus is relatively brief, typically occurring between February and April. By the time the males were introduced on May 15th, the ewes were entering a more shallows anestrus (or transition period), as they were preparing for the upcoming breeding season, as typically occurs in adult ewes. This timing was carefully considered to align with the natural reproductive cycle, ensuring that the ewes were in an optimal physiological state for the male effect to take place. This new information has been added in the Material and Methods section.
• Complete coordinates are missing: longitude, altitude
Response: coordinates have been added
• Describe: photoperiod of the region (and daylight hours during the study period)
Response: More information about seasonal variation in daylight hours and temperatures have been added
• Maximum and minimum temperatures
Response: more information on temperature variations have been addedd
• Mention the name of the grasses and shrubs that the sheep consumed.
Response: We have listed the main species of grasses and shrubs consumed by the sheep.
• Were the groups separated? At what distance?
Response: the four group were kept together (see response to comment L123 of the Rev3 for more details: “... To clarify, the 200 ewes were divided into four experimental groups of 50 ewes each, but they were all kept together as a single flock. The 10 rams were introduced into this combined flock of 200 ewes, so there were 10 rams for the entire flock, maintaining a ratio of 1 ram per 20 ewes (10/200 = 1/20). Each group of 50 ewes was part of this larger flock, but the rams were not specifically assigned to individual groups.”
• Management of the males: did they alternate between groups? What body condition did they have?
Response: The rams involved in the study had an Body Condition Score (BCS) > 3.5. These details have now been added to the manuscript for clarity and completeness.
Once introduced to the group of ewes, they remained with them for the entire 50-day period without any rotation between groups. The rams independently managed the reproductive activity of the entire flock during this time.
• Was their diet not increased over time even when the sheep were pregnant?
Response: As specified in the Materials and Methods section, the supplementation consisted of a fixed amount of 400 g of concentrate feed (crude protein 20.4% and 12.5 MJ ME/kg DM) per day. This quantity remained constant throughout the study period.
• In the Materials and Methods they do not clearly describe that the females were milking (how many days does the milking period or the suckling period of the calves last?) and what their production was, although in lines 310-312 they talk about not finding a difference in production between groups but they do not report that result.
Response: Material and Methods section has been implemented with a more detailed description of the milking and suckling periods and explicitly reporting the production results to address this oversight.
• Why is there such a difference in BCS in the same herd? Grazing habits?, hierarchies, less time after weaning their calves?
Response: Differences of up to 1.5 in BCS within the same herd are not uncommon and can indeed be attributed to a variety of factors, including those mentioned by the reviewer. Grazing habits, social hierarchies, and time since weaning are all well-documented influences on BCS variation. Our study aimed precisely to investigate these differences and associate them with reproductive performance. By examining these variations, we sought to provide a more comprehensive understanding of how individual BCS fluctuations within a herd can impact reproductive outcomes, particularly in the context of the Mediterranean region and its unique production systems.
• A significant number of twins were not recorded, not even in the high BCS groups?
Response: The Sarda sheep breed is characterized by inherently low prolificacy. Additionally, farmers in this region have historically discouraged twinning, as they prefer raising single lambs. This preference stems from management strategies aimed at optimizing lamb survival and maternal health. As a result, ewe lambs born from twin births are typically not retained as replacements to avoid introducing the genetic predisposition for twinning into their flocks. This cultural and genetic management practice likely explains the limited number of recorded twin births, even among ewes with high BCS.
• It is desirable to report the data of the Pregnancy Diagnosis at 45 and 90 days to have a clearer idea of some reproductive parameters.
Response: A sentence has been added to the manuscript (lines 291-292) providing a clearer understanding of the reproductive parameters.
• Reporting the levels of insulin, leptin IGF-1 would support the results of this study
Response: Thank you for your suggestion. Our future research plans include measuring hormonal levels such as insulin, leptin, and IGF-1 to further clarify the mechanisms between BCS and reproductive activity. However, these measurements were not included in the current study.
• Why was prolificacy low in all groups?
Response: as stated above, the Sarda breed is inherently known for its low prolificacy, which is likely the main factor explaining the observed low prolificacy across all groups in the study.
It is desirable to include if there are records of:
Return to estrus
Ovulatory rate,
Pregnant females
Females that aborted (in which period; 1st, 2nd or 3rd third of gestation).
Weight of the offspring
Weight of the litter
Types of birth (single, twins, triplets) between groups
Response: Thank you for your suggestions. Some of the requested data are already included in the manuscript. For instance, information on the return to estrus is addressed in lines 331–334; while the difference between the number of ewes diagnosed as pregnant and those that lambed is now reported in Lines 291-292. No abortions were observed during the last trimester of pregnancy. Litter size within each group is provided in Table 2, and it is worth noting that no triplets were observed in the study. Unfortunately, lamb birth weights were not recorded.
We hope this clarification addresses your concerns, and we are open to further suggestions.
DISCUSION
They discuss a very extensive paragraph (lines 292-315) on milk production and do not report this parameter
Response: The average milk production level has now been included in the manuscript (lines 452–453).
Table 2. Empty: how many days into the study?
Response: The Table 2 report the number of ewes that remained empty at the end of the observations, which was 200 days after the introduction of the rams.
Table 6. It would be appropriate to refer to the result (very outdated) of 68.34 in group BCS 3.5, T50
Response: Thank you for pointing this out. The table contained an error, which has now been corrected.
Miscellaneous
Line
26 ”The male effect is commonly used in sheep farming as an alternative to hormonal treatments, but its success can be influenced by factors such as BCS”. Missing bibliographical reference
Response: citations are not included in the abstract. A new sentence has been added (Lines 67-69) supported by references within Introduction section.
15 and 83 Hormone free (are they referring to exogenous hormones?)
Response: the sentences have been corrected adding “exogenous”
151 meaning of DRIL
Response: the sentence has been deleted
268. Says Bronson (1998) (24)
Response: the error has been corrected.
352 The surname is Viñoles
Response: the surname has been corrected.
The identification of the bars in figure 1 is not clearly visible (next to line 219).
Response: Thank you for your feedback. The figure that was difficult to read has been replaced with a new version, which we hope will be clearer.
Reviewer 3 Report
Comments and Suggestions for Authors
I have carefully read the manuscript presented by Luridiana et al. I find it well structured and well written.
The topic addressed is certainly of interest, of great relevance and falls perfectly within the scope of the journal.
The experimental design is appropriate to the objectives of the study. The experimental samples are sufficient to obtain solid results. The conclusions are largely supported by the obtained results.
I would like to point out only a few small changes to be made before considering the manuscript for publication:
L 64: insert a brief explanation in brackets for flushing.
L 67: first cite all the factors that negatively influence this reproductive management technique before writing “inadequate response”.
L 104: inappropriate citation, here as elsewhere, cite the original source and not who reporting it.
L 107: also in this case cite the source. Furthermore, the productions must be referred to a specific milking period.
L 108: specify that now you are writing about the experimental subjects.
L109: the term “food” is used throughout the text, I recommend replacing it with “feed”, more appropriate for livestock.
L 123: I don’t understand how the 10 rams were divided into the 4 experimental groups to obtain a ratio of 1/20. Specify or correct.
L 144: specify the model.
L 148 – 150: repeated sentence. Delete.
L 158 – 160: leave only “(version 4.4. 0, R 158 Core Team (2024”). Move the rest to the bibliography section.
L 165: random factors are usually indicated with a lowercase letter.
L 174: delete the repetition.
Author Response
Rev3
I have carefully read the manuscript presented by Luridiana et al. I find it well structured and well written.
The topic addressed is certainly of interest, of great relevance and falls perfectly within the scope of the journal.
The experimental design is appropriate to the objectives of the study. The experimental samples are sufficient to obtain solid results. The conclusions are largely supported by the obtained results.
Response: Thank you for your valuable feedback. Your suggestion has been carefully addressed as outlined below.
I would like to point out only a few small changes to be made before considering the manuscript for publication:
L 64: insert a brief explanation in brackets for flushing.
Response: a brief explanation for flushing has been added in brackets
L 67: first cite all the factors that negatively influence this reproductive management technique before writing “inadequate response”.
Response: the sentence has been revised.
L 104: inappropriate citation, here as elsewhere, cite the original source and not who reporting it.
Response. We appreciate the comment. However, we confirm that reference [20] is indeed the original source for the statement regarding the Sarda sheep breed. This information is explicitly stated in the "Materials and Methods" section of the cited work: “Sarda is the most farmed and the best Italian breed of dairy sheep, and Sardinia (with more than 3 million of farmed ewes) has the highest density of sheep in Italy. This breed produces the highest amount of sheep milk in Italy and a considerable quantity of sheep milk at world level.” As no additional references or sources are cited within the original text, we believe that reference [20] is correctly used in this context.
L 107: also in this case cite the source. Furthermore, the productions must be referred to a specific milking period.
Response: the production data reported in our study specifically refer to a defined milking period. These data are drawn from our extensive research experience in the field of dairy sheep, which spans over 30 years. In our
L 108: specify that now you are writing about the experimental subjects.
Response: the sentence has been revised.
L109: the term “food” is used throughout the text, I recommend replacing it with “feed”, more appropriate for livestock.
Response: the term “food” has been corrected in “feed” here and throughout the text
L 123: I don’t understand how the 10 rams were divided into the 4 experimental groups to obtain a ratio of 1/20. Specify or correct.
Response: Thank you for your comment. To clarify, the 200 ewes were divided into four experimental groups of 50 ewes each, but they were all kept together as a single flock. The 10 rams were introduced into this combined flock of 200 ewes, so there were 10 rams for the entire flock, maintaining a ratio of 1 ram per 20 ewes (10/200 = 1/20). Each group of 50 ewes was part of this larger flock, but the rams were not specifically assigned to individual groups.
We hope this clarification addresses your concern, and we are happy to provide further explanation if necessary.
L 144: specify the model.
Response: The spectrophotometer model has been specified in the manuscript.
L 148 – 150: repeated sentence. Delete.
Response. Repeated sentence removed.
L 158 – 160: leave only “(version 4.4. 0, R 158 Core Team (2024”). Move the rest to the bibliography section.
Response: citation has been revised. Additionally, the numbering of all subsequent entries has been corrected accordingly.
L 165: random factors are usually indicated with a lowercase letter.
Response: The random factors have now been indicated with lowercase letters.
L 174: delete the repetition.
Response: “(Bates et al., 2015)” removed.
Reviewer 4 Report
Comments and Suggestions for Authors
This manuscript describes a study of 200 Sarda dairy ewes, describing reproductive success (i.e., fertility rate, no. of ewes lambing, litter size, and no. of days from ram introduction to lambing) and metabolic parameters (i.e., blood glucose, blood cholesterol, blood triglycerides, blood total proteins, and blood albumin) during the breeding period based on initial body condition scoring. However, information about the ewes and their management is missing from the manuscript and as a result, important information for the reader is absent. It is recommended to provide additional context in the introduction, and elaborate on their results/findings/trends in the discussion to better connect literary sources to experimental design to results to implications. These changes will enhance clarity to the reader.
Simple Summary: Authors should add age range of the sheep as they did in the abstract.
Abstract: Please include that the BCS scoring system is on a 1 – 5 scale for added clarity.
Introduction: Additional background information about the metabolic parameters [used in this study] as they relate to reproductive performance in livestock would be beneficial for reader understanding.
Materials and Methods: Please add more details on the nutrition regime for these ewes. Were they flushed and how long were they on the concentrate? Also consider adding weight ranges for each body condition (if measured). Did days since previous lambing influence DRIL? Was genetic background similar across BCS groups. What else could have influenced results in this study and what are additional limitations to this study?
Discussion: The background information about the metabolic parameters that were measured in this study is useful, though the discussion as it relates to the study findings across groups is lacking. There is quite a bit of longitudinal measures, though those trends were not included in the discussion.
Line 24: Insert “dairy” for readers unfamiliar with the breed (“Sarda dairy breed ewes”).
Lines 120 – 122: Authors need to how scorers were trained and how many scorers were used.
Lines 129 – 132: Authors need to describe who was assessing behavior and how the sexual behavior was assessed.
Lines 142 – 144: For study replication purposes, please provide the name of the commercial kits and references that they are accurate in dairy sheep.
Lines 151 – 155: These sentences are repeated from lines 148 – 151, with some additional information.
Lines 360 – 361: Please describe why the present study’s findings contrast those of Caldeira et al.
Table 3/Figure 1. I believe DRIL information and the information in table 3 would be better represented graphically using a bar graph and not lumping lambing dates into 10-day periods (unlike Figure 1 which is not useful with the 10-day periods).
Author Response
Rev 4 Round1
This manuscript describes a study of 200 Sarda dairy ewes, describing reproductive success (i.e., fertility rate, no. of ewes lambing, litter size, and no. of days from ram introduction to lambing) and metabolic parameters (i.e., blood glucose, blood cholesterol, blood triglycerides, blood total proteins, and blood albumin) during the breeding period based on initial body condition scoring. However, information about the ewes and their management is missing from the manuscript and as a result, important information for the reader is absent. It is recommended to provide additional context in the introduction and elaborate on their results/findings/trends in the discussion to better connect literary sources to experimental design to results to implications. These changes will enhance clarity to the reader.
Response: Thank you for your valuable feedback. Your suggestion has been carefully addressed as outlined below.
Simple Summary: Authors should add age range of the sheep as they did in the abstract.
Response: the age range of the enrolled ewes has been added (Line 16).
Abstract: Please include that the BCS scoring system is on a 1 – 5 scale for added clarity.
Response: the specification about BCS scoring system on a 1–5 scale has been added (Line 28-29)
Introduction: Additional background information about the metabolic parameters [used in this study] as they relate to reproductive performance in livestock would be beneficial for reader understanding.
Response. Some information has been added in the Introduction about the relationship between metabolic parameters and reproductive performance (Lines 89-96)
Materials and Methods: Please add more details on the nutrition regime for these ewes. Were they flushed and how long were they on the concentrate? Also consider adding weight ranges for each body condition (if measured). Did days since previous lambing influence DRIL? Was genetic background similar across BCS groups. What else could have influenced results in this study and what are additional limitations to this study?
Response:
- more information about flushing has been added in Lines 140-149;
- weight ranges have been added in Lines 153-156;
- The influence of days since previous lambing on reproductive efficiency (DRIL) was not evaluated in this study. Specifically, one of the selection criteria for the ewes included in the study was that they had lambed between November 1st and December 1st of the previous season (line 151-152), to ensure consistency in management and minimize variations due to the inter-lambing interval.
- A statement regarding the genetic background homogeneity among the BCS groups has been added to the Materials and Methods section (Lines 158-162).
- A short section about factor potentially influencing the study has been added (Lines 195-199); additionally, a brief mention in the discussion regarding the potential limitations of the study has already been included in the first round of the revision process (Lines 476-479)
Discussion: The background information about the metabolic parameters that were measured in this study is useful, though the discussion as it relates to the study findings across groups is lacking. There is quite a bit of longitudinal measures, though those trends were not included in the discussion.
Response: Thank you for your valuable feedback. In response to your comment, we have revised the Discussion to include considerations regarding the metabolic parameters measured in the study and their trends across groups. These additions aim to better contextualize the findings and address the potential links between metabolic status and reproductive performance. We hope these changes address your concerns and enhance the manuscript.
Line 24: Insert “dairy” for readers unfamiliar with the breed (“Sarda dairy breed ewes”).
Response: done
Lines 120 – 122: Authors need to how scorers were trained and how many scorers were used.
Response: scoring was performed by the veterinary team who care for the flock. This specification has been added at Line 166-167. Furthermore it was already specified in lines 210-211 that the surveys were always carried out by the same evaluators.
Lines 129 – 132: Authors need to describe who was assessing behavior and how the sexual behavior was assessed.
Response: the observation regarding the sexual behaviour of the rams were made by the farm staff who are responsible for the daily management of the animals. (specification added at Lines 190-191). As stated in Lines 191-192 the “Observed behaviors included anogenital sniffing, nudging, and mounting attempts directed at the ewes.”
Lines 142 – 144: For study replication purposes, please provide the name of the commercial kits and references that they are accurate in dairy sheep.
Response: name of the commercial kits has already been included in the first round of the revision process.
Lines 151 – 155: These sentences are repeated from lines 148 – 151, with some additional information.
Response: Thank you for pointing this out. This mistake was identified and corrected during the first round of revisions, and the repeated sentence has been removed.
Lines 360 – 361: Please describe why the present study’s findings contrast those of Caldeira et al.
Response: Thank you for pointing out this inaccuracy. We have carefully reviewed the data and clarified the comparison with the findings of Caldeira et al. in the revised manuscript. Specifically, we have corrected the statement regarding total cholesterol, emphasizing that Caldeira et al. focused on total lipids, which showed no significant differences between groups, and did not explicitly report trends for total cholesterol. We have also elaborated on the potential reasons for discrepancies between the two studies to ensure a more accurate and transparent discussion (Lines 466-470)
Table 3/Figure 1. I believe DRIL information and the information in table 3 would be better represented graphically using a bar graph and not lumping lambing dates into 10-day periods (unlike Figure 1 which is not useful with the 10-day periods).
Response: Table 3 has been removed and replaced with a bar graph as suggested.
Round 2
Reviewer 2 Report
Comments and Suggestions for Authors
I wish to acknowledge the work of the authors as the quality of the document has improved.
1. Between lines 72 and 78 there are two paragraphs with the same text
2. In lines 151 I suggest the terminology 1 = emaciated, 5 = obese
3. In the section on Materials and Methods, the following is not yet clear:
Were the groups separated? At what distance?
I suggest that the following response from the authors can be used for the text in the document
Response: the four group were kept together (see response to comment L123 of the Rev3 for more details: “... To clarify, the 200 ewes were divided into four experimental groups of 50 ewes each, but they were all kept together as a single flock. The 10 rams were introduced into this combined flock of 200 ewes, so there were 10 rams for the entire flock, maintaining a ratio of 1 ram per 20 ewes (10/200 = 1/20). Each group of 50 ewes was part of this larger flock, but the rams were not specifically assigned to individual groups.”
4. The text on the lines 388-389 Says: In the present study no significant differences emerged in the production levels of the ewes within each group or between groups (average daily milk production 1800±180 g/die). It seems that the result of an evaluated variable is being reported, however I think that it refers to the state that the females were in when the experimental groups were formed, then, that this data should be included in the paragraph between lines 141-144.
5. It is somewhat difficult to follow the answers since the number of lines referred to is not real in the document
Author Response
Rev 2 Round2
I wish to acknowledge the work of the authors as the quality of the document has improved.
Response: Thank you for your kind acknowledgment and positive feedback. We greatly appreciate your valuable suggestions, which have significantly contributed to improving the quality of the manuscript.
1. Between lines 72 and 78 there are two paragraphs with the same text
Response 1.: Thank you for pointing out this error. The repeated text between lines 72 and 78 has been removed.
2. In lines 151 I suggest the terminology 1 = emaciated, 5 = obese
Response 2.: The terminology "1 = emaciated, 5 = obese" has been added to clarify the BCS scale in the corresponding section.
3. In the section on Materials and Methods, the following is not yet clear:
Were the groups separated? At what distance?
Response 3: In the Materials and Methods section, reference is made to different types of separation, which may have caused some confusion. The 200 selected ewes, divided into four experimental groups based on BCS (2.0, 2.5, 3.0, and 3.5), were kept together as a single group, but they were isolated from the rest of the flock. The rams, on the other hand, were kept 5 km away from the ewes to prevent any exposure to sight, sound, or smell during the isolation period. This separation was essential to ensure the male effect could be properly established when the rams were introduced later.
I suggest that the following response from the authors can be used for the text in the document
Response: the four group were kept together (see response to comment L123 of the Rev3 for more details: “... To clarify, the 200 ewes were divided into four experimental groups of 50 ewes each, but they were all kept together as a single flock. The 10 rams were introduced into this combined flock of 200 ewes, so there were 10 rams for the entire flock, maintaining a ratio of 1 ram per 20 ewes (10/200 = 1/20). Each group of 50 ewes was part of this larger flock, but the rams were not specifically assigned to individual groups.”
Response: The response to the reviewer has been integrated into the section above, as suggested.
4. The text on the lines 388-389 Says: In the present study no significant differences emerged in the production levels of the ewes within each group or between groups (average daily milk production 1800±180 g/die). It seems that the result of an evaluated variable is being reported, however I think that it refers to the state that the females were in when the experimental groups were formed, then, that this data should be included in the paragraph between lines 141-144.
Response 4: Thank you for your valuable observation. We have moved the sentence regarding the milk production levels of the ewes to lines 176-177, as suggested, where it better fits the description of the initial state of the females before the experimental groups were formed.
5. It is somewhat difficult to follow the answers since the number of lines referred to is not real in the document
Response 5.: Thank you for your comment. During the various revisions of the manuscript, we also found it challenging to pinpoint the exact line numbers. We have made every effort to indicate the changes as accurately as possible in this revision, and we hope the references to the line numbers are now clearer.